# Malnutrition drives infection susceptibility and dysregulated myelopoiesis that persists after refeeding intervention

Alisa Sukhina[1,2], Clemence Queriault[1], Saptarshi Roy[1], Elise Hall[1], Kelly Rome[1], Muskaan Aggarwal[2], Elizabeth Nunn[3], Ashley Weiss[1,2], Janet Nguyen[1], F Chris Bennett[4,5], Will Bailis[1,2]*

[1]Department of Pathology and Laboratory Medicine, Children's Hospital of Philadelphia, Philadelphia, United States; [2]Department of Pathology and Laboratory Medicine, Perelman School of Medicine, University of Pennsylvania, Philadelphia, United States; [3]Department of Physiology, Perelman School of Medicine at the University of Pennsylvania, Philadelphia, United States; [4]Department of Psychiatry, Perelman School of Medicine , University of Pennsylvania, Philadelphia, United States; [5]Division of Neurology, Children's Hospital of Philadelphia, Philadelphia, United States

*For correspondence:
bailisw@chop.edu

## eLife Assessment

This **important** work advances our understanding of the impact of malnutrition on hematopoiesis and subsequently infection susceptibility. Support for the overall claims is **convincing** in some respects and incomplete in terms of identifying mechanism as highlighted by reviewers. This work will be of general interest to those in the fields of hematopoiesis, malnutrition, and dietary influence on immunity.

**Abstract** Undernutrition remains a major global health crisis, with nearly 1 billion people experiencing severe food insecurity. Malnourished individuals are especially vulnerable to infectious diseases, which is the leading cause of morbidity and mortality for this population. Despite the known link between undernutrition and infection susceptibility, the mechanisms remain poorly understood, and it is unclear whether refeeding can reverse nutritionally acquired immunodeficiency. Here, we investigate how malnutrition leads to immune dysfunction and the ability of refeeding to repair it. Malnourished mice show an inability to control sublethal *Listeria monocytogenes* infection, reduced immune cell function and expansion, and early contraction before pathogen clearance. Myelopoiesis is particularly affected, with fewer neutrophils and monocytes present both before and after infection in malnourished mice. While refeeding restores body mass, lymphoid organ cellularity, and T cell responses, refed mice remain susceptible to *Listeria* infection, revealing that recovery from lymphoid atrophy alone is not sufficient to restore protective immunity. Accordingly, peripheral neutrophils and monocytes fail to fully recover, and emergency myelopoiesis remains impaired in refed animals. Altogether, this work identifies dysregulated myelopoiesis as a link between prior nutritional state and immunocompetency, indicating that food scarcity is an immunologic risk factor, even after nutritional recovery.

## Introduction

Almost 500 million adults and over 200 million children are affected by undernutrition worldwide, with over half of all childhood deaths linked to undernutrition (*United Nations, 2021*; *UNICEF, 1998*). In the past 5 years, these numbers worsened due to economic instability ushered by COVID-19 pandemic, and they are projected to further increase throughout the impending climate crisis (*Mbow et al., 2019*; *United Nations, 2022*). In patients with undernutrition, the largest contributor to morbidity and mortality is infectious disease (*Fan et al., 2022*; *Rice et al., 2000*). It has been demonstrated that undernutrition alone puts patients, especially children, at a higher risk of developing long-lasting disability or dying from an infection (*Antwi, 2011*; *Bhargava, 2016*; *Bourke et al., 2016*; *Caulfield et al., 2004*; *Rice et al., 2000*; *Schlaudecker et al., 2011*; *Sinha et al., 2021*). In addition to increased susceptibility to infection, chronically undernourished individuals have been reported to have impaired barrier function, atrophy of immune organs, and less effective responses to vaccines (*Bhattacharjee et al., 2021*; *Bourke et al., 2016*; *Collins and Belkaid, 2022*; *Prendergast, 2015*). In keeping with this, undernutrition is the leading cause of secondary immunodeficiency in the world (*Chinen and Shearer, 2010*). Considering the geographical overlap between areas with high prevalence of undernutrition and infectious disease and the persistent nature of the global undernutrition crisis, it is critical to investigate how undernutrition causes immunodeficiency and contributes to poor infection outcomes (*Mbow et al., 2019*; *Murray and Lopez, 1996*; *Pelletier, 1994*; *Roberts, 2017*; *Rohr et al., 2019*; *Sinha et al., 2021*; *UNICEF-WHO-World Bank JME Working Group, 2021*).

Despite the first link between undernutrition and poor infection outcomes being posited over a century ago, the cellular mechanisms underlying undernutrition-induced immunodeficiency remain poorly resolved (*Bourke et al., 2019*; *Bourke et al., 2016*; *Hess, 1932*; *Newsholme, 1908*). The primary explanation given for this dysfunction has centered on defects in lymphocyte biology. Patient data and animal studies suggest that T, B, and NK cells are reduced in the periphery of patients with undernutrition and experimental animals (*Campbell et al., 2020*; *Cason et al., 1986*; *Contreras et al., 2018*; *Howard et al., 1999*; *Nájera et al., 2004*; *Saha et al., 1977*; *Schattner et al., 1990*; *Yang et al., 2009*). Moreover, animal models of short-term fasting and protein deficiency have been found to impair T cell expansion, cytokine production, and memory recall responses (*Chatraw et al., 2008*; *Iyer et al., 2012*; *Mengheri et al., 1992*; *Procaccini et al., 2012*; *Saucillo et al., 2014*; *Taylor et al., 2013*). While the contribution of lymphocyte dysfunction to nutritionally acquired immunodeficiency is well established, the impact of prolonged undernutrition on other immune cell populations and the role they play in disease susceptibility is not well understood.

Prior studies have reported disparate impacts of undernutrition on immunity and infection, with some finding dietary or caloric restriction enhances inflammation or immunity, while others observing a loss in immune function (*Bhattacharjee et al., 2021*; *Campbell et al., 2020*; *Chatraw et al., 2008*; *Collins et al., 2019*; *Han et al., 2023*; *Hasegawa et al., 2012*; *Iyer et al., 2012*; *Palma et al., 2021*; *Pena-Cruz et al., 1989*; *Piccio et al., 2008*; *Starr et al., 2016*; *Sun et al., 2001*; *Taylor et al., 2013*). These discrepancies likely result from whether animals are subjected to caloric restriction alone or if there is also corresponding limitation in key micro- and macronutrients and whether the restriction in food intake is administered as a short-term fast or is sustained (*Cerqueira and Kowaltowski, 2010*; *Contreras et al., 2018*; *Meydani et al., 2016*; *Palma et al., 2021*; *Yan et al., 2021*). To this end, caloric and nutrient restriction diets significantly differ in weight loss patterns, physiology, and nutritional state (*Cerqueira and Kowaltowski, 2010*). While both diets are associated with reduced inflammation, caloric restriction is understood to support both lifespan and health span, in stark contrast to chronic undernutrition (*Collins and Belkaid, 2022*; *Contreras et al., 2018*; *Goldberg et al., 2015*; *Green et al., 2022*; *Hasegawa et al., 2012*; *Howard et al., 1999*; *Jordan et al., 2019*; *Piccio et al., 2008*; *Spadaro et al., 2022*; *Sun et al., 2001*; *Yang et al., 2009*).

Beyond our mechanistic understanding of nutritionally acquired immunodeficiency, there is limited knowledge on whether this dysfunction is reversible (*Contreras et al., 2018*; *Schattner et al., 1990*; *Vaisman et al., 2004*). Because current treatment guidelines for patients with undernutrition utilize refeeding protocols that focus on weight gain as the indicator of recovery, the effects of refeeding on the immune system are understudied (*Ashworth et al., 2003*). Moreover, whether prior exposure to malnutrition has durable effects on the ability to control infection following refeeding remains unknown.

Here, we investigate the impact of nutritionally acquired immunodeficiency on the immune responses to *Listeria monocytogenes* infection and the ability of refeeding intervention to restore immune competency. We employ a chronic murine dietary restriction model and demonstrate that it effectively recapitulates the hallmarks of human undernutrition. We find that chronic undernutrition results in a specific atrophy of immune organs that is not mirrored in essential organs, such as the liver and kidney. This loss of lymphoid tissue is accompanied by a broad reduction in both innate and adaptive immune cell compartments. Undernourished mice fail to control sublethal *L. monocytogenes* infection, either succumbing to disease or failing to clear the pathogen long term. We find that while these mice undergo the initial phase of immune cell expansion following infection, they fail to sustain it, and the response rapidly undergoes contraction. Accordingly, we observe that T cells in these mice display muted expansion, accelerated contraction, and impaired effector function. We further find that undernutrition selectively impairs steady-state and emergency myelopoiesis, resulting in reduced neutrophil and monocyte abundance prior to and post-infection. Finally, we demonstrate that while refeeding protocols are sufficient to restore body mass, growth, and reverse global lymphoid atrophy, refed mice remain more susceptible to infection even months after recovering size and peripheral immune cell numbers. In contrast to the ability of refeeding to reverse the effects of undernutrition on lymphoid numbers, we go on to show refed mice display impaired emergency myelopoiesis as well as neutrophil and monocyte abundance. Altogether, our work identifies dysregulated myelopoiesis as a major factor contributing to increased susceptibility to infection during chronic undernutrition. Furthermore, we for the first time demonstrate that refeeding protocols are not sufficient to reverse defects in the ability of mice to control infection or dysregulated myelopoiesis, despite outwardly displaying a recovery from malnutrition. We believe these findings have important implications for global public health policy and medical standards of care not only for treating people actively experiencing chronic malnutrition, but also for individuals who have experienced and recovered from food scarcity as part of their life history.

## Results

### Sustained dietary restriction recapitulates the hallmarks of nutritionally acquired immunodeficiency

To investigate the relationship of undernutrition and immunodeficiency, we began by adopting a faithful model of chronic malnutrition. In patients, undernutrition is associated with (1) a consistent weight loss of 10% or more compared to the age-appropriate body weight that occurs in under 6 months and persists for months to years; (2) stunting of growth; and (3) a change in body composition associated with weight loss (*Institute of Medicine, 2000*; *Maleta, 2006*; *Martins et al., 2011*). While short-term fasting and macronutrient or caloric restriction models have been previously used, we elected to employ a 40% reduced diet (40RD) model, which provides both the weight loss as well as micro- and macronutrient deficiency over extended periods of time (*Cerqueira and Kowaltowski, 2010*). For the 40RD system, the average food intake of C57Bl6 mice was measured at baseline consumption. After a week of baseline measurements, mice were randomly assigned to either the ad libitum fed (AL) or the 40RD group. 40% of the average food intake by weight was removed from the diet of 40RD mice (*Figure 1a*). Animal weight, body length, and body condition score (BCS) were then regularly surveyed to measure wasting, stunting, and body composition, respectively. Highlighting the ability of the model to reliably generate a state of moderate undernutrition, 40RD mice consistently lost 10% of their initial body weight (IBW) in 4 weeks (*Figure 1b*) and achieved an average 20% loss of IBW relative to the AL group (*Figure 1c*). At the same time, BCSs decreased significantly in 40RD mice compared to the AL group over the course of 4 weeks (*Figure 1d*). This decrease in weight and conditioning was further accompanied by delayed growth and stunting in 40RD mice, as measured by body length (*Figure 1e*).

Beyond these gross changes in body condition, patients with undernutrition also exhibit severe lymphoid organ atrophy (*Beisel, 1996*; *Bourke et al., 2016*). In keeping with this, we observed a reduction in spleen, thymus, and lymph node size and mass in mice given the 40RD diet, while organs such as the liver and kidney were unchanged, suggesting a specific effect of malnutrition on the immune system (*Figure 1f*). We observed a corresponding reduction in the cellularity of the spleen and thymus, with a more modest loss of cellularity observed in the bone marrow (*Figure 1g*). Collectively,

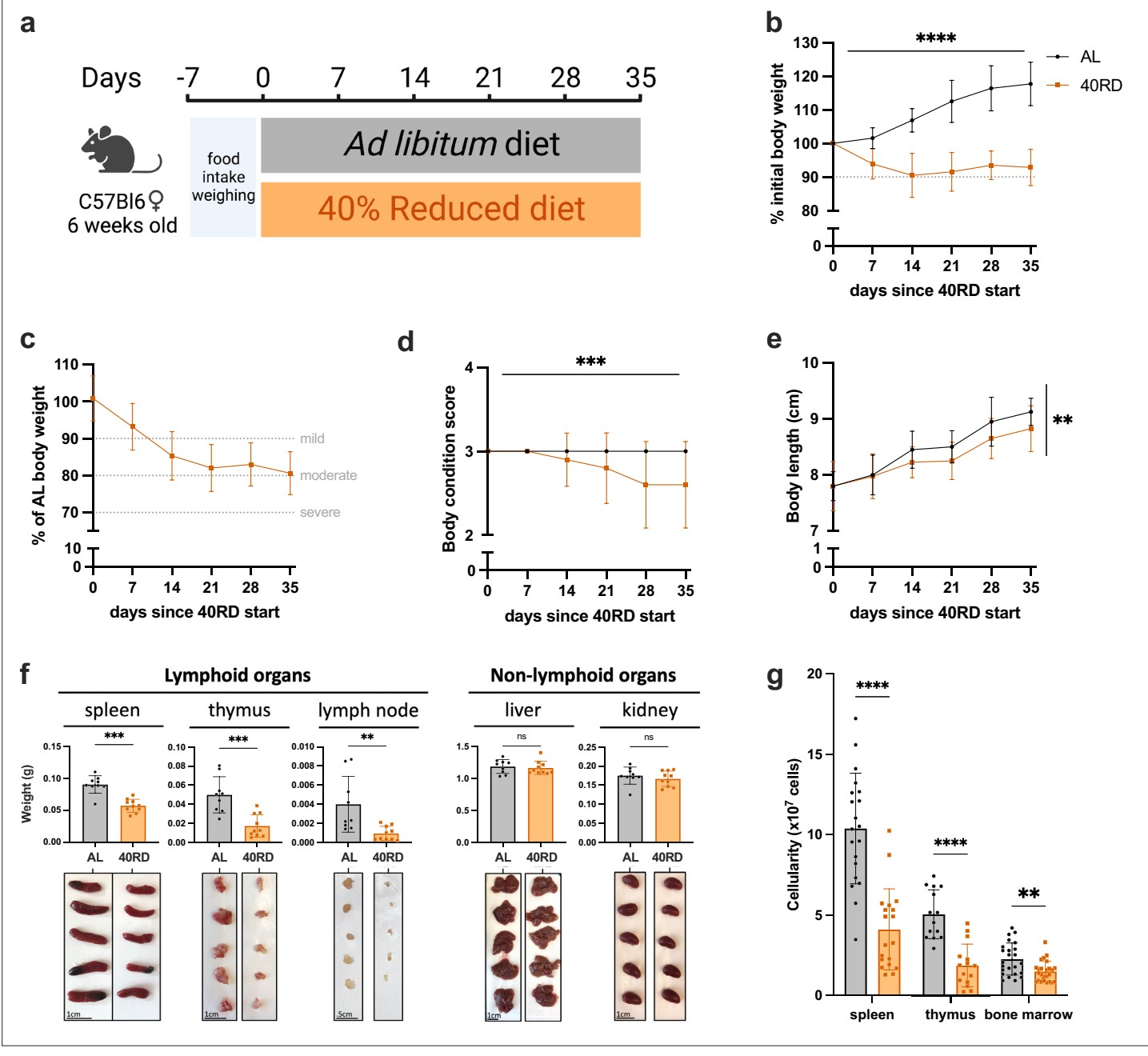

**Figure 1.** Sustained dietary restriction recapitulates the hallmarks of nutritionally acquired immunodeficiency. (**a**) Schematic of experimental design for 40% reduced diet (40RD, orange) in comparison to control ad libitum (AL, black) diet. (**b**) Body weight of AL and 40RD mice as a percentage of initial body weight over time (*n* = 50). The dotted line represents 10% of initial body weight lost. (**c**) Body weight of 40RD mice as a percentage of age-matched average AL body weight over time (*n* = 50). Each dotted line represents clinical designations of undernutrition severity. (**d**) Body condition score of AL and 40RD mice over time (*n* = 10). (**e**) Body length of AL and 40RD mice over time, measured from the nose tip to the base of the tail (*n* = 10). (**f**) Comparative weights of AL and 40RD lymphoid and non-lymphoid tissues (*n* = 10) with representative photos of the corresponding organs. Scale bars 1 cm (0.5 cm for lymph nodes). (**g**) Total live cell counts for whole spleen (*n* = 15), thymus (*n* = 15), and bone marrow (*n* = 10). Statistics: (**b–g**) Plotted as mean ± SD; (**b, d**) simple linear regression with slope comparisons; (**e**) simple linear regression with elevation comparison; and (**f, g**) two-tailed Mann–Whitney test. ** P ≤ 0.01, *** P ≤ 0.001, **** P ≤ 0.0001.

The online version of this article includes the following source data for figure 1:

**Source data 1.** Raw numerical values for *Figure 1* plots.

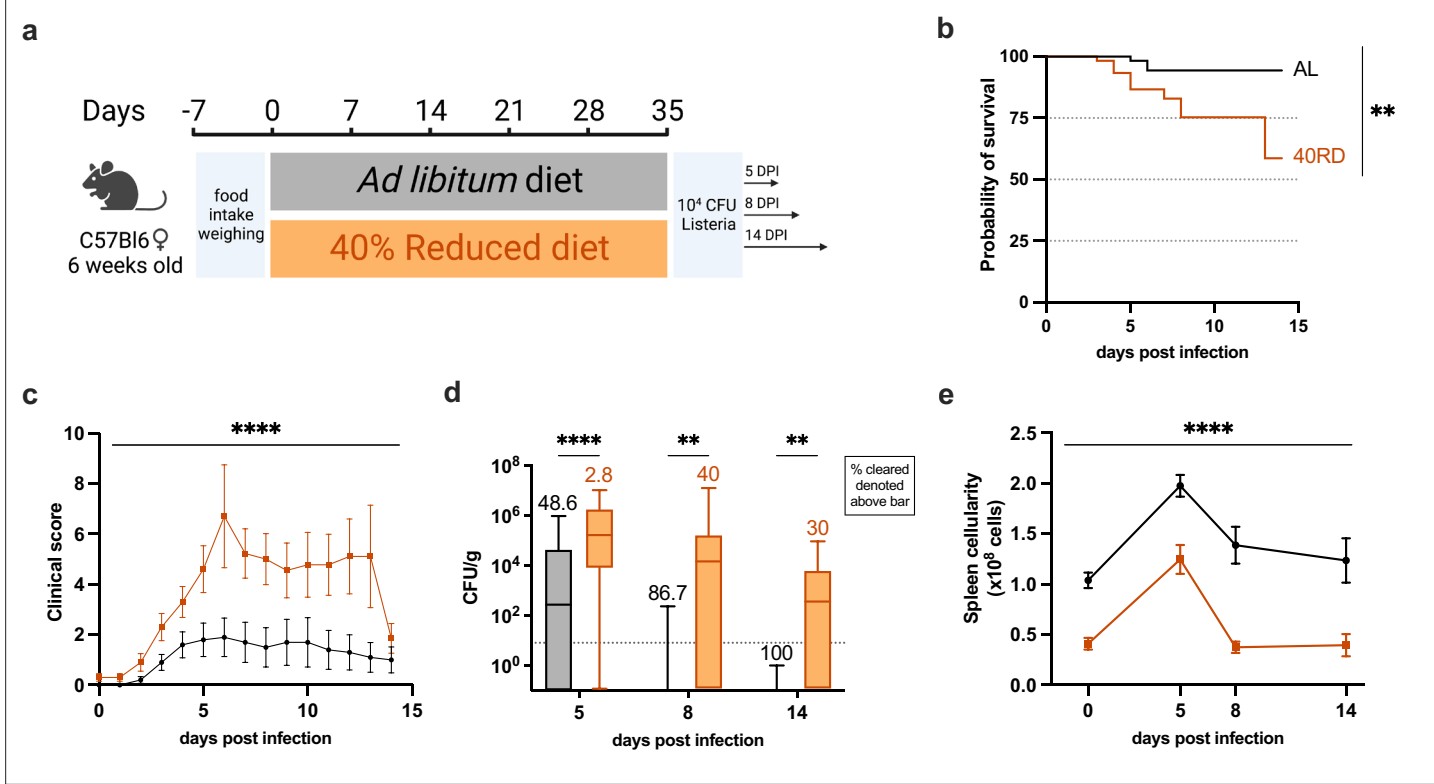

**Figure 2.** Chronic malnutrition results in a failure to control sublethal *L. monocytogenes* infection. (**a**) Schematic of Lm infection (10⁴ CFUs per mouse) experimental design in AL (orange) and 40RD (black) mice. Mice were maintained on the corresponding diet throughout the course of the infection. (**b**) Probability of survival for infected AL and 40RD mice over time. The curves represent pooled data from three experimental groups: 5DPI (*n* = 25), 8DPI (*n* = 15), and 14DPI (*n* = 10). Statistics done via log-rank test. (**c**) Clinical score for infected AL and 40RD mice over time from the 14DPI group. Plotted as mean ± SEM; statistics done via mixed-effect two-way ANOVA analysis. (**d**) Pathogen burden in liver tissue of AL and 40RD mice. Percentage of mice that cleared the pathogen on a given day is represented as numbers above corresponding bars. The dotted line represents the limit of detection. Plotted as box and min to max whiskers; statistics done via two-tailed Mann–Whitney test for each time point. (**e**) Total splenocyte counts for infected AL and 40RD mice over time. Uninfected spleen cell counts are the same as used for *Figure 1g*. Plotted as mean ± SEM; statistics done via mixed-effect two-way ANOVA analysis. ** P ≤ 0.01, **** P ≤ 0.0001.

The online version of this article includes the following source data and figure supplement(s) for figure 2:

**Source data 1.** Raw numerical values for *Figure 2* plots.

**Figure supplement 1.** AL and 40RD mice were infected with 10⁴ CFUs of *Listeria monocytogenes* per mouse.

**Figure supplement 1—source data 1.** Raw numerical values for *Figure 2—figure supplement 1* plots.

our findings demonstrate that the 40RD model reproduces the lymphoid atrophy found in chronically ill patients with undernutrition as well as other models of undernutrition in rodents (*Beisel, 1996*; *Bourke et al., 2016*; *Cason et al., 1986*; *Contreras et al., 2018*; *Yang et al., 2009*).

## Chronic malnutrition results in a failure to control sublethal *L. monocytogenes* infection

We next sought to investigate how malnutrition impacted immune responses to infection. While the link between poor infection outcomes and undernutrition has been extensively documented, the immunologic mechanisms resulting in poor infection resolution remain unknown (*Bourke et al., 2016*; *Bourke et al., 2019*; *Dubos, 1955*; *Ishikawa et al., 2012*; *Rice et al., 2000*). To address this, we turned to the ovalbumin-expressing *Listeria monocytogenes* (Lm-Ova) system, a classic model of bacterial infection that permits the tracking of adaptive immunity, innate immunity, and bacterial burden. Both 40RD and AL mice were infected with a sublethal dose of Lm-Ova, and then infection progression, pathogen clearance, and immune responses were assessed at 5, 8, and 14 days post-infection (*Figure 2a*). Whereas all mice in the AL group survived, nearly half of the 40RD mice

became moribund and required euthanasia (*Figure 2b*). Even among surviving mice, the 40RD group displayed higher clinical scores than the AL group throughout the course of infection (*Figure 2c*). Consistent with this, 40RD mice failed to clear bacteria at the rate of the AL group, with some mice exhibiting persistent infection (*Figure 2d*). Among the AL group, 48.6% and 86.7% of the mice were able to clear bacteria by day 5 and 8 post-infection, respectively, with all mice resolving infection by day 14. In contrast, 40RD mice exhibited delayed clearance kinetics, with only a third of the mice becoming pathogen-free by day 14 (*Figure 2d*).

These observations prompted us to ask whether 40RD mice could mount a normal immune response to infection. To begin addressing this, we examined immune cell expansion in AL and 40RD mice throughout the course of infection. We found that while splenocyte numbers increased at early time points following infection in 40RD mice, the magnitude and duration of this expansion were significantly lower compared to AL controls (*Figure 2e*). Thus, chronic malnutrition is sufficient to not only induce immune atrophy but further abrogates the capacity of mice to mount and sustain an immune response to infection.

## Chronic malnutrition diminishes T cell expansion and function while accelerating contraction during infection

Considering the broad defects observed in infected, chronically undernourished mice, we next aimed to investigate the effects malnutrition had on specific immune cell populations during this response.

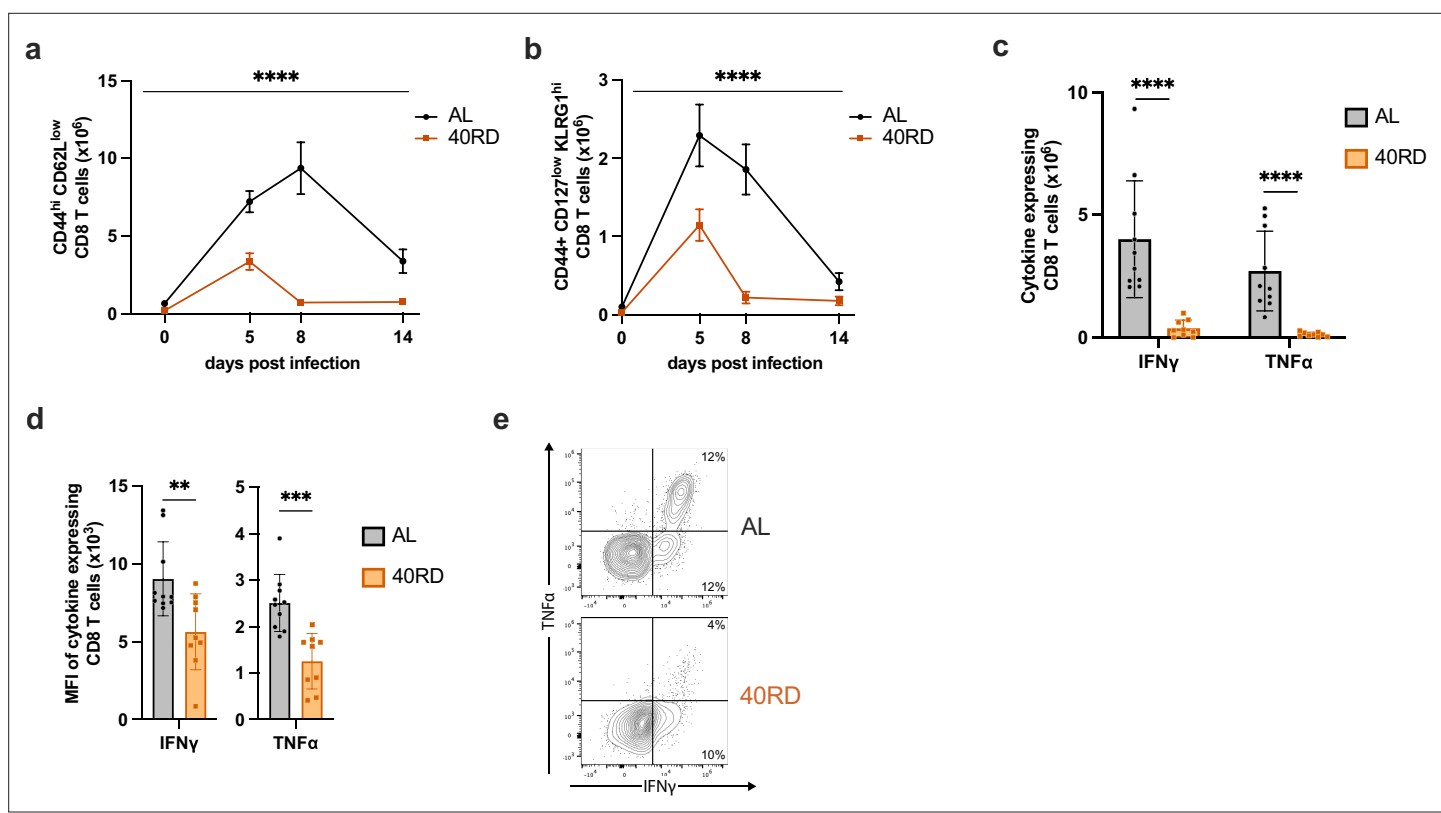

**Figure 3.** Chronic malnutrition diminishes T cell expansion and function while accelerating contraction during infection. AL and 40RD mice were infected with Lm at $10^4$ CFUs per mouse. Splenocytes were counted, and flow cytometry was performed at days 0, 5, 8, and 14 post-infection to evaluate the total cell number of (**a**) antigen-experienced CD8[+] T cells gated on live single CD8[+] CD44hi CD62Llow and (**b**) short-lived effector cells further gated on KLRG1hi CD127low. Plotted as mean ± SEM; statistics done via mixed-effect two-way ANOVA analysis. Splenocytes from AL (*n* = 10) and 40RD (*n* = 10) mice were harvested at day 8 post-infection and stimulated ex vivo with OVA peptide for 6 hr. Intracellular flow cytometry was performed to quantify (**c**) total cell number of and (**d**) mean fluorescence intensity (MFI) of antigen-experienced CD8[+] T cells expressing IFNγ and TNFα. Plotted as mean ± SD; statistics done via two-tailed Mann–Whitney test for each cytokine. (**e**) Representative flow cytometry data of (**c, d**), with average frequencies shown. ** P ≤ 0.01, *** P ≤ 0.001, **** P ≤ 0.0001.

The online version of this article includes the following source data for figure 3:

**Source data 1.** Raw numerical values for *Figure 3* plots.

As malnourished patients are known to exhibit a reduction in lymphocyte numbers, we assessed whether 40RD mice exhibited signs of lymphopenia following infection (*Cason et al., 1986*; *Nájera et al., 2004*; *Saha et al., 1977*). We observed that both prior to and after infection, 40RD mice displayed reduced numbers of B cells, CD4 T cells, and CD8 T cells compared to controls (*Figure 2—figure supplement 1*). Despite this overall loss in lymphocyte number, the relative frequency of each population was either unchanged or elevated, indicating that while malnutrition leads to a global reduction in immune cell numbers, lymphocytes are less impacted than other immune cell populations (*Figure 2—figure supplement 1*).

Considering the role CD8 T cells play in resolving intracellular bacterial infections and the failure of 40RD mice to clear infection at later time points, we next examined whether the kinetics of T cell expansion and contraction were similarly impacted. To do so, we tracked T cell function throughout the course of infection, on day 5 (early response), day 8 (peak response), and day 14 (contraction phase) post-infection (*Qiu et al., 2018*; *Zenewicz and Shen, 2007*). We found that the antigen-experienced (CD44$^+$) and short-lived effector (CD127$^{low}$KLRG1$^{high}$) CD8$^+$ T cell numbers were significantly reduced throughout the infection (*Figure 3a, b*). These effector-like populations expanded minimally during early and peak response and contracted quickly by day 14, prior to pathogen clearance in 40RD mice. In addition to these defects in expansion/contraction dynamics, we further tested whether chronic malnutrition also impaired T cell function. We thus evaluated both the frequency and per-cell cytokine-producing capacity of T cells at day 8 post-infection. We observed that T cells from undernourished mice displayed both a decreased frequency of cytokine-producing cells and a decreased capacity for producing cytokines on a per-cell basis (*Figure 3c–e*). Altogether, these findings suggest that impaired T cell function along with insufficient T cell expansion and premature contraction prior to pathogen clearance contribute to the failure of chronically undernourished mice in controlling *L. monocytogenes* infection.

## Chronically malnourished mice display dysregulated myelopoiesis

The observation that 40RD mice exhibited elevated pathogen burden at early time points prior to the peak of the adaptive immune response between days 5 and 8 prompted us to investigate whether innate immune responses were similarly impaired. While total bone marrow cellularity was equivalent between the 40RD and AL groups, examination of specific innate immune populations in bone marrow revealed that steady-state neutrophil and monocyte levels were significantly lower in undernourished mice, both in absolute numbers and relative abundance (*Figures 1g and 4a–d*; *Figure 4—figure supplement 1*). In keeping with this, we observed that splenic neutrophil and monocyte abundance were significantly lower in 40RD mice at steady state and underwent impaired expansion following infection, with splenic monocyte frequency elevated at steady state and comparable to control mice post-infection (*Figure 4b–d*; *Figure 4—figure supplement 1*). These steady-state changes could be observed early after mice were placed on the 40RD diet, with total peripheral blood cellularity and neutrophil abundance significantly reduced by 1 week post-dietary restriction (*Figure 4—figure supplement 1e, f*).

Neutrophil and monocyte production is maintained in the bone marrow during steady-state and through emergency myelopoiesis in both bone marrow and spleen upon infection (*Janssen et al., 2016*; *Kim, 2010*; *Manz and Boettcher, 2014*; *Yvan-Charvet and Ng, 2019*). We thus evaluated how undernutrition affected the key myeloid progenitor populations, pre-granulocyte/monocytes (pre-GM) and granulocyte/monocyte progenitors (GMP) (*Figure 4e*). Consistent with a loss of mature neutrophils and monocytes, we found that myelopoiesis was significantly impaired in undernourished mice. Within the bone marrow, pre-GM and GMP cells were present at lower numbers in steady-state 40RD mice (*Figure 4f, g*). Upon infection, these bone marrow progenitor populations were capable of undergoing expansion, with pre-GM cells failing to reach AL levels, whereas GMP cells reached comparable numbers (*Figure 4f–h*). These defects were more exaggerated in the spleen, where 40RD mice displayed diminished numbers and frequency of pre-GM and GMP cells both pre- and post-infection (*Figure 4f–h*). Altogether, we find that undernutrition impairs steady-state myelopoiesis and extramedullary emergency myelopoiesis, blunting the innate immune response against a bacterial pathogen.

## Refeeding intervention reverses wasting, stunting, and global immune atrophy

One of the primary strategies taken to support patients with undernutrition with health complications, including infections, is to refeed them by slowly increasing caloric intake to levels expected for their

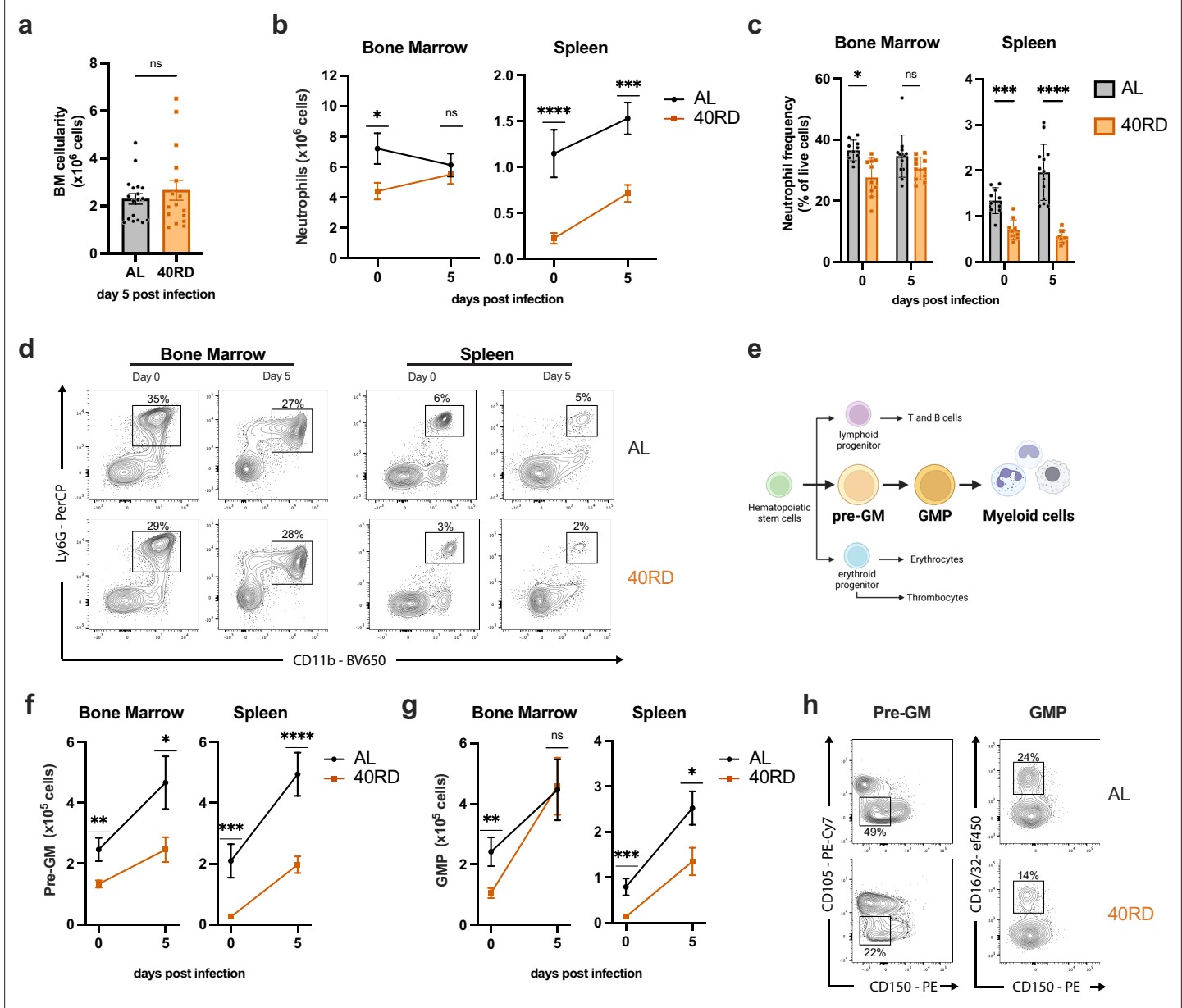

**Figure 4.** Chronically malnourished mice display dysregulated myelopoiesis. (**a**) Total bone marrow cell counts from day 5 post-infection AL (*n* = 18) and 40RD (*n* = 16) mice. Bone marrow cells and splenocytes were counted, and flow cytometry was performed at days 0 (*n* = 10 for both AL and 40RD) and 5 post Lm infection in AL and 40 RD mice to evaluate (**b**) the total cell number of neutrophils and (**c**) the relative frequency of neutrophils among live cells. (**d**) Representative flow cytometry data for the results in (**b, c**), with average frequencies shown. (**e**) A simplified schematic representation of myelopoiesis showing pre-GM and granulocyte/monocyte progenitor (GMP) as key progenitors in granulocyte/monocyte lineage. Bone marrow cells and splenocytes were counted, and flow cytometry was performed at days 0 and 5 post Lm infection in AL and 40 RD mice to evaluate the total cell number of (**f**) pre-GM cells (Lineage⁻ Sca1⁻ CD117⁺ CD150⁻ CD16/32⁻ CD105⁻) and (**g**) GMP cells (Lineage⁻ Sca1⁻ CD117⁺ CD150⁻ CD16/32⁺). (**h**) Representative flow cytometry results for the myeloid progenitor data in (**f, g**), with average frequencies shown. Statistics: (**a–c, f, g**) Plotted as mean ± SEM; statistics done via two-tailed Mann–Whitney test for each time point. * P ≤ 0.05, ** P ≤ 0.01, *** P ≤ 0.001, **** P ≤ 0.0001.

The online version of this article includes the following source data and figure supplement(s) for figure 4:

**Source data 1.** Raw numerical values for *Figure 4* plots.

**Figure supplement 1.** AL and 40RD mice were infected with 10⁴ CFUs of *Listeria monocytogenes* per mouse.

**Figure supplement 1—source data 1.** Raw numerical values for *Figure 4—figure supplement 1* plots.

age group (*Ashworth et al., 2003*). Despite the widespread employment, the effects of this intervention on immune dysfunction are not known. To address this, we developed a refeeding protocol that safely reintroduces ad libitum feeding to 40RD mice to test whether defects in immune responses can be rescued through weight gain and nutrient availability. Undernourished mice undergoing a refeeding protocol (RF) were maintained on a standard 40RD diet for 4 weeks or until 10% BWL. Then, the RF mice were given 10% extra food by weight every 2 days for a week. On the last day of the refeeding protocol, the RF mice were given ad libitum access to feed. Mice were maintained for an additional 6–8 weeks on the ad libitum feed until they reached a normal weight range for their age (*Figure 5a*). The RF group was able to regain IBW during the refeed period and continued to gain weight at an accelerated pace for the first month (*Figure 5b*). After the initial increase in weight, the weight gain pace of RF mice slowed down to the same rate as the AL group (*Figure 5c*). We further observed that while RF mice were able to recover growth, the mice maintained a modest but significantly shorter body length than AL controls for their age (*Figure 5d*), in keeping with persistent stunting common in patients with undernutrition (*Ashworth et al., 2003*).

With the refeeding protocol established, we set out to test whether restoring body mass was sufficient to reverse the lymphoid atrophy found in 40RD mice. Paralleling recovery in weight, we found that cellularity in the spleen, thymus, and bone marrow were comparable between AL and RF groups (*Figure 5e*). Moreover, we observed that the weight of lymphoid organs has returned to the AL levels in the RF group (*Figure 5f*). Together, these findings indicate that refeeding intervention is sufficient to restore global lymphoid atrophy.

## Refeeding intervention fails to restore immunocompetency and normal myelopoiesis

After observing successful reversal of the lymphoid atrophy in RF mice, we tested whether refeeding would be sufficient to restore their ability to control *L. monocytogenes* infection. To test this, we infected age-matched AL, 40RD, and RF mice with a sublethal dose of Lm-OVA (*Figure 6a*). While refeeding was able to limit morbidity compared to 40RD mice, a portion of RF animals succumbed to infection, whereas all AL mice remained viable (*Figure 5b*). Consistent with this, we observed that mice with ongoing malnutrition maintained the highest pathogen burdens and that RF mice were unable to resolve infection with the same kinetics as AL mice, with less than half of RF mice clearing bacteria (*Figure 5c*). These data indicate that despite refeeding being sufficient to reverse malnutrition-induced lymphoid atrophy, prior exposure to chronic undernutrition leads to persistent susceptibility to infection.

Having observed this durable impairment in immune protection, we next aimed to identify the specific facets of malnutrition-induced immune dysfunction that failed to recover in RF mice. In keeping with the steady-state recovery observed, post-infection splenic and bone marrow cellularity were equivalent between RF and AL mice (*Figure 6d*). Moreover, we found that T and B cell numbers were restored in RF mice, and all lymphocyte populations underwent comparable expansion to AL controls after infection (*Figure 6—figure supplement 1*). Similarly, we observed no differences in the number of antigen-experienced T cells in the spleen nor T cell functional capacity (*Figure 6e–g*). Thus, refeeding permits recovery of lymphocyte number, lymphocyte expansion capacity, and T cell function in response to infection, suggesting lymphocyte dysfunction is unlikely to be responsible for the persistent susceptibility observed.

In contrast to the restoration of the lymphocyte response, we observed sustained defects in the myeloid compartment. RF mice displayed impaired peripheral expansion of neutrophil and monocyte populations after infection, both by frequency and number, with neutrophil number diminished in steady-state RF mice as well (*Figure 6i, j*; *Figure 6—figure supplement 2a, b*). Within the bone marrow, we observed a reduced frequency in steady-state monocyte and neutrophil abundance as well as reduced total neutrophil numbers; however, both populations expanded upon infection, reaching control numbers (*Figure 6—figure supplement 2c–f*). Having seen this, we next asked whether RF mice exhibited a persistent impairment in either central or emergency myelopoiesis. Whereas myeloid progenitor populations were not perturbed in the bone marrow, we found that RF mice exhibited significantly reduced frequency and numbers of splenic pre-GM and GMP cells following infection, with the GMP population also diminished in steady-state mice (*Figure 6j–l*; *Figure 6—figure supplement 3*). Altogether, our findings demonstrate that refeeding intervention uncouples a global

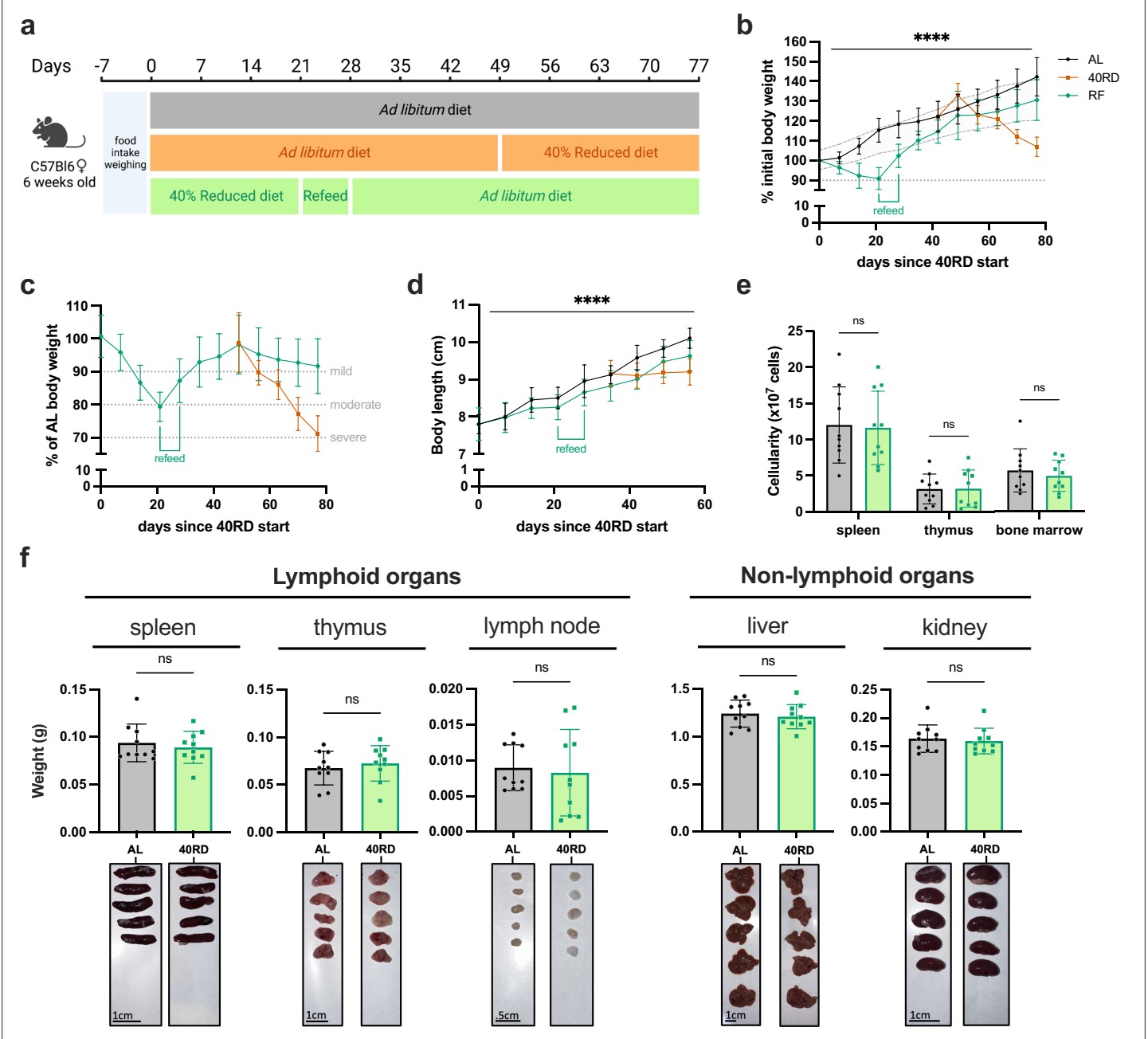

**Figure 5.** Refeeding intervention reverses wasting, stunting, and global immune atrophy. (**a**) Schematic of the experimental design for refeeding intervention (RF) in comparison to age-matched 40RD and control AL diet. (**b**) Body weight of AL (*n* = 25), 40RD (*n* = 10), and RF (*n* = 25) mice as a percentage of their initial body weight over time. The dotted line represents 10% of initial body weight lost, and the shaded area represents the normal weight range for age-matched female C57Bl6 mice (c). (**c**) Body weight of 40RD (*n* = 10) and RF (*n* = 25) mice as a percentage of age-matched average AL body weight over time. Each dotted line represents clinical designations of undernutrition severity. (**d**) Body length of AL and 40RD mice over time, measured from the nose tip to the base of the tail (*n* = 10). (**e**) Total cell counts for whole spleen, thymus, and bone marrow from AL and RF mice (*n* = 10). (**f**) Comparative weights of AL and RF lymphoid and non-lymphoid tissues (*n* = 10) with representative photos of the corresponding organs. Scale bars 1 cm (0.5 cm for lymph nodes). Statistics: (**b–f**) Plotted as mean ± SD; (**b, d**) simple linear regression with slope comparisons; and (**e, f**) two-tailed Mann–Whitney test. **** P ≤ 0.0001.

The online version of this article includes the following source data for figure 5:

**Source data 1.** Raw numerical values for *Figure 5* plots.

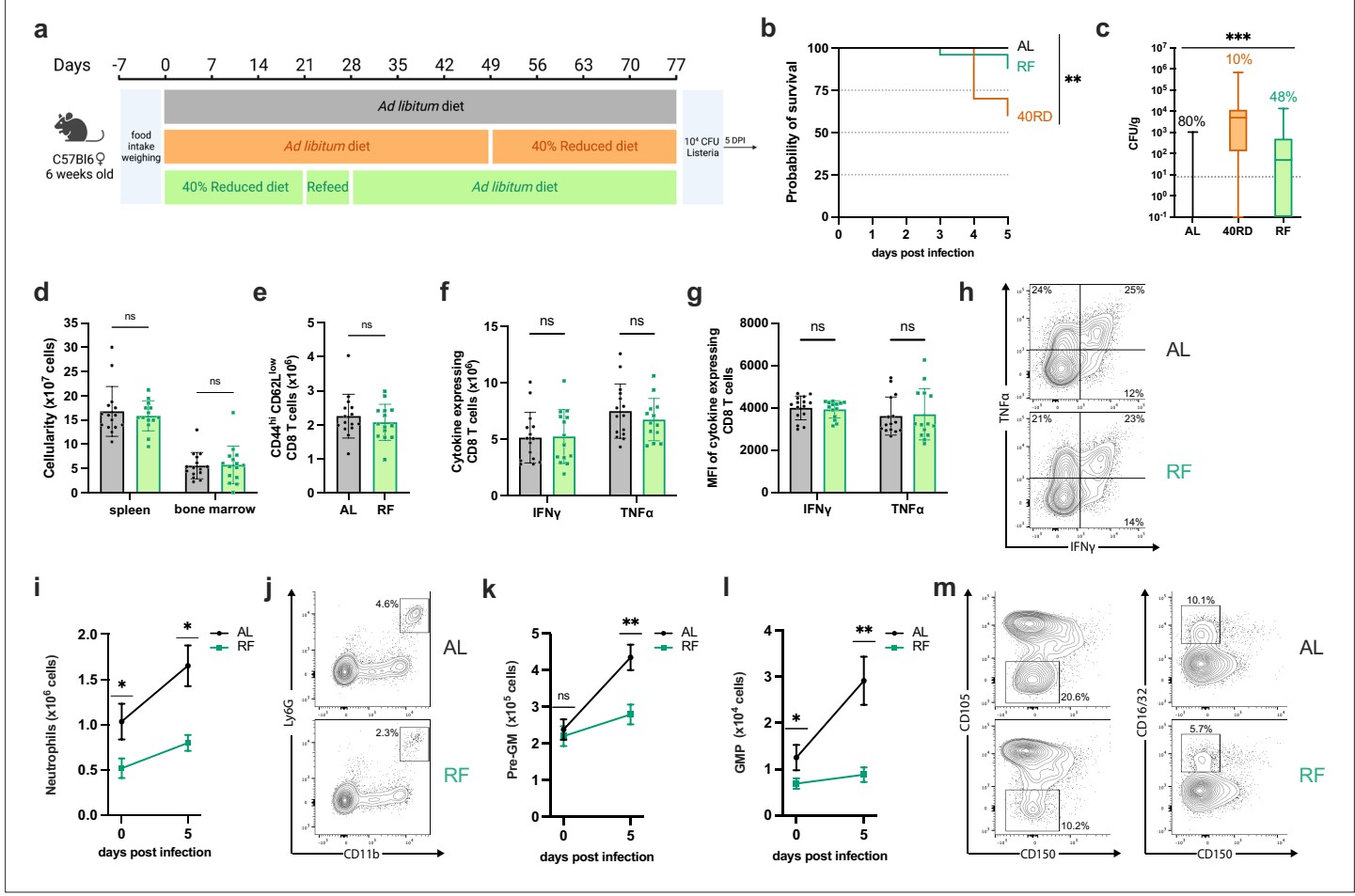

**Figure 6.** Refeeding intervention fails to restore immunocompetency and normal myelopoiesis. (**a**) Schematic of Lm infection ($10^4$ CFUs per mouse) experimental outline in AL, 40RD, and RF mice. Mice were maintained on the corresponding diet throughout the course of the infection. (**b**) Probability of survival for infected AL ($n = 25$), 40RD ($n = 10$), and RF ($n = 25$) mice over time. Statistics done via log-rank test. (**c**) Pathogen burden in liver tissue of five DPI AL ($n = 25$), 40RD ($n = 10$), and RF ($n = 25$) mice. Percentage of mice that cleared the pathogen on a given day is represented as numbers above corresponding bars. The dotted line represents the limit of detection. Plotted as box and min to max whiskers; statistics done via Kruskal–Wallis test. (**d**) Total splenocyte and bone marrow cell counts for AL ($n = 15$) and RF ($n = 15$) mice at day 5 post-infection. (**e**) Total cell number of CD44+CD8+ T cells in AL and RF mice at days 0 and 5 post-infection. Splenocytes from AL and RF mice were harvested at day 5 post-infection and stimulated ex vivo with OVA peptide for 6 hr. Intracellular flow cytometry was performed to quantify (**f**) total cell number of and (**g**) mean fluorescence intensity (MFI) of antigen-experienced CD8+ T cells expressing IFNγ and TNFα. (**h**) Representative flow cytometry results for data in (**f, g**), with average frequencies shown. (**i**) Total splenic neutrophils abundance in AL and RF mice at days 0 and 5 post-infection. (**j**) Representative flow cytometry results for day 5 post-infection neutrophil data in (**i**), with average frequencies shown. At days 0 and 5 post-infection, spleens from AL and RF were evaluated for the total number of (**k**) pre-GM cells and (**l**) granulocyte/monocyte progenitor (GMP) cells. (**m**) Representative flow cytometry results plots for the myeloid progenitor data in (**k, l**), with the average frequencies shown. Statistics: (**d–g**) Plotted as mean ± SD; statistics done via two-tailed Mann–Whitney test for each category. (**i, k, l**) Plotted as mean ± SEM; statistics done via two-tailed Mann–Whitney test for each time point. * $P \le 0.05$, ** $P \le 0.01$, *** $P \le 0.001$.

The online version of this article includes the following source data and figure supplement(s) for figure 6:

**Source data 1.** Raw numerical values for *Figure 6* plots.

**Figure supplement 1.** AL and RF mice were infected with $10^4$ CFUs of *Listeria monocytogenes* per mouse.

**Figure supplement 1—source data 1.** Raw numerical values for *Figure 6—figure supplement 1* plots.

**Figure supplement 2.** AL and RF mice were infected with $10^4$ CFUs of *Listeria monocytogenes* per mouse.

**Figure supplement 2—source data 1.** Raw numerical values for *Figure 6—figure supplement 2* plots.

**Figure supplement 3.** AL and RF mice were infected with $10^4$ CFUs of *Listeria monocytogenes* per mouse.

**Figure supplement 3—source data 1.** Raw numerical values for *Figure 6—figure supplement 3* plots.

recovery in lymphoid atrophy from an enduring impairment in emergency myelopoiesis, resulting in lasting susceptibility to infection.

## Discussion

Here, we employ a murine model of chronic undernutrition that phenocopies many of the hallmarks of human undernutrition (stunting, wasting, and changes in body composition) to characterize how chronic malnutrition leads to nutritionally acquired immunodeficiency. We demonstrate that sustained malnutrition results in global lymphoid atrophy, with loss of both innate and adaptive immune cell populations. Additionally, we identified a unique and severe impairment in neutrophil abundance and myelopoiesis, with a similar but more modest effect on monocytes. Our data suggest that the combination of diminished homeostatic levels of these critical myeloid populations along with impaired emergency myelopoiesis leads to poor control of early *L. monocytogenes* infection, while a blunted T cell response to infection and their premature contraction results in lack of effective infection resolution. We then tested whether a refeeding intervention could restore protective immunity to infection. We found that restoring food access permitted mice to regain weight and size as well as reverse many signs of immunodeficiency. Refeeding was sufficient to recover lymphoid organ atrophy, the abundance of most circulating lymphocytes, and T cell cytokine production. Nonetheless, refed mice continued to display increased susceptibility to *L. monocytogenes* infection along with sustained defects in neutrophil and monocyte abundance and emergency myelopoiesis. These data demonstrate that exposure to chronic undernutrition can result in lasting changes to specific compartments of the immune system, with long-term implications for the health of the individual even after recovering growth and size.

Previous studies have suggested that impaired pathogen responses during undernutrition are driven by lymphocyte loss and functional impairment (*Beisel, 1996*; *Beisel, 1982*; *Institute of Medicine, 2000*). Low levels of circulating lymphocytes have been well documented in patients with undernutrition and various restrictive diet models (*Campbell et al., 2020*; *Cason et al., 1986*; *Contreras et al., 2018*; *Howard et al., 1999*; *Saha et al., 1977*; *Schattner et al., 1990*; *Yang et al., 2009*). Our data from the refeeding model suggest that while these defects likely contribute to poor resolution of infection, they are not sufficient to explain why previously malnourished mice remain vulnerable to infection, as these mice recover from lymphoid atrophy and display normal T cell responses. Nonetheless, we find consistent with other groups that undernourished mice exhibit a steady-state reduction in circulating T cells. Upon infection, the T cell compartment is capable of undergoing expansion in malnourished conditions, but the magnitude of this response is blunted and quickly contracts prior to pathogen clearance. We also found that there was reduced abundance of antigen-experienced T cells throughout the infection. Additionally, we found that the intrinsic ability of T cells to produce inflammatory cytokines was reduced. In keeping with work demonstrating that T cells from patients with undernutrition display reduced function, we find that T cells from malnourished mice have impaired cytokine-producing capacity post-infection. Altogether, our work supports clinical observations and earlier studies delineating the relationship of undernutrition on lymphoid atrophy and T cell dysfunction and has revealed the reversible nature of these defects after refeeding intervention.

In contrast to the recovery found in the adaptive immune cells after refeeding, our work demonstrates that exposure to chronic periods of malnutrition results in durable defects in the myeloid compartment. We find that undernourished mice display impaired myelopoiesis and a loss in abundance in critical peripheral myeloid populations that does not properly recover upon refeeding. Indeed, some small cohort studies have shown that circulating neutrophil numbers are reduced in undernourished children and that hematopoiesis is impacted in anorexic patients (*Kohama et al., 2021*; *Vaisman et al., 2004*). Thus, while hematopoiesis and peripheral immune cell numbers are broadly sensitive to an animal's ongoing nutritional status, specific subsets are uniquely sensitive to periods of undernutrition and exhibit nutritional scarring that uncouples their activity from future improvements in dietary intake. These findings raise the possibility that lasting epigenetic changes within the myeloid compartment occur during exposure to undernutrition, akin to what has been observed for trained immunity (*Netea et al., 2020*). Indeed, a recent study in a small cohort of undernourished children reported changes in H3 acetylation in peripheral blood mononuclear cells (*Kupkova et al., 2021*).

While this work provides a novel link between myelopoiesis and nutritional state, many questions remain. Our study investigated the effect of chronic undernutrition during *L. monocytogenes* infection;

however, the extent to which these findings apply more broadly to other models of bacterial infection or innate inflammation is unclear. Moreover, our malnutrition model broadly restricted food intake without regard to the components of diet that are required for immunocompetency nor consideration for potential interactions of the magnitude of food restriction with immunodeficiency and the rate of onset. It will be of interest in future studies to resolve the extent to which the phenotypes observed are due to the gross loss of calories, specific macronutrients, certain micronutrients, and the grade of dietary restriction enforced. In addition, our study focused on the effects of dietary restriction on adult, normal weight mice. Whether similar phenotypes would emerge if obese animals were dietarily restricted and if weight loss was achieved through pharmacologic means (e.g. GLP-1 targeted drugs) is not clear. Likewise, how similar dietary restriction might affect the immune system if it were initiated in juvenile mice is not addressed here. Mechanistically, additional work is needed to address whether the changes in myelopoiesis are due to changes within the hematopoietic compartment, non-hematopoietic cells, and/or to systemic levels of key cytokines and hormones – such as G-CSF, interferons, and leptin – both in malnourished and refed mice. Finally, the known impact dietary restriction has on microbiota composition and in turn how this might durably impact the immune system is unaccounted for in this work and warrants investigation (*Gordon et al., 2012*; *Blanton et al., 2016*).

Altogether, our work offers new insight into the contribution of myelopoiesis in nutritionally acquired immunodeficiency and the ability of refeeding intervention to treat these defects. The burden of morbidity and mortality of infections on undernourished individuals is one of the leading global health crises, and evidence of impaired vaccine responses in undernourished people is putting in question the world's ability to protect the most vulnerable populations with its most reliable strategies for prevention (*Bhattacharjee et al., 2021*; *Bhargava, 2016*; *Bourke et al., 2019*; *Bourke et al., 2016*; *Dubos, 1955*; *Ishikawa et al., 2012*; *Prendergast, 2015*; *Rice et al., 2000*; *Saha et al., 1977*). We provide evidence that even prior exposure to food scarcity may be sufficient to permanently alter future immune responses. This suggests that the current dietary status of an individual may be insufficient information to understand the impact nutrition has on their immune system. Future work delineating the specific components of diet required to sustain immune health in the face of food scarcity will be critical for developing nutritional interventions or broader food supplementation programs for at-risk populations. We believe that this work indicates periods of poverty and food insecurity may be an important factor in patient medical history as well as for formulating global health policy in these areas.

## Materials and methods
### Mice and diets
All mice were female on a C57Bl6 background maintained at Children's Hospital of Philadelphia (CHOP) animal facility. Mice were either purchased from Jackson laboratories at 6 weeks of age and acclimated in the CHOP facility for 1 week or bred in-house from breeders also obtained from Jackson laboratories. All 40RD and RF experiments were performed with mice between 6 and 8 weeks of age. Mice were housed in groups of four to five animals for all experiments. All experiments were performed on age-matched littermate controls. All mice were maintained on LabDiet 5001 from weaning/arrival to facility.

This study was performed in strict accordance with the recommendations in the Guide for the Care and Use of Laboratory Animals of the National Institutes of Health and with PHS-approved animal welfare (assurance number D16-00280). All of the animals were handled according to approved Institutional Animal Care and Use Committee (IACUC) protocols (#001325) of the Children's Hospital of Philadelphia, and issues of pain, distress, and injury were addressed throughout this process.

For the 40RD diet, mice were maintained on an ad libitum diet until the start of the experiment. One week before the start of the experiment, average food intake was measured per 5 mice per day. Average food consumption for all experimental animals between 6 and 8 weeks of age was 4 g/day/mouse. At the start of the experiment, mice were weighed to make sure that all the animals were in the range of ±1.5 g and randomly assigned to an AL or a 40RD group. AL mice were maintained as before with unlimited access to chow. 40RD mice were placed on the 40% reduced diet by weight, resulting in 2.4 g/day/mouse of feed. Consistent weight loss and no competition for food was observed in 40RD mice. 40RD mice lost 10% of their IBW every 4 weeks. If mice lost over 20% of their IBW, they

were euthanized. During the 40RD diet, experimental animals did not exhibit behavioral or clinical changes and only moderate changes in body composition and growth stunting. When all 40RD mice reached 10–15% IBW loss, they were considered undernourished and utilized for further experiments.

For the RF diet, mice were set up in the identical way to the 40RD diet until they reached 10–15% IBW loss. At which point they underwent a refeeding intervention developed using Guidelines for the inpatient treatment of severely malnourished children. The refeeding intervention lasted a week and increased the weight of available food to the restricted mice by 10% every 2 days. On the 40RD diet, mice had access to 2.4 g/day/mouse (60% of the initial average food consumption). On day 1 of the refeeding intervention, mice had access to 2.8 g/day/mouse (70%). On day 3, they had access to 3.2 g/day/mouse (80%). On day 5, it was 3.6 g/day/mouse (90%). On day 7, mice were given unrestricted food access. RF mice were then maintained on ad libitum feed for a month after they entered normal weight range for their age,and their food intake normalized to the AL age-matched controls. At that point, mice were considered refed and were utilized for further experiments.

40RD mice and mice on the 1-week-long refeeding intervention were fed daily between 4 pm and 6 pm to avoid interference with circadian rhythm. Mice on the 40RD and intervention consumed all food provided to them between feeding windows. All mice were observed daily for signs of sickness. All mice were weighed every 2–3 days for three times a week total. Food intake for all mice was recorded either daily, if fed daily, or every 2–3 days for three times a week total if fed ad libitum. Mice were maintained on assigned diets during infection and sepsis experiments. Mice had unlimited access to drinkable water.

## Body length, clinical score, and BCS

For 30 40RD mice and 30 age-matched AL controls, body length and BCS were recorded weekly. For 25 RF mice and 25 age-matched AL controls, body length was recorded weekly. Body length was recorded using a standard centimeter ruler from the base of the tail to the tip of the nose. The length was recorded in 0.25 cm increments. BCS was recorded using IACUC Pain and Distress Recognition Policy guidelines in a blinded manner by the same observer for all time points. BCS was rated on the scale of 1–5, as follows: (1) emaciated, prominent skeletal structure and distinct vertebrae segmentation; (2) underconditioned, segmentation of vertebrae, dorsal pelvic bones palpable; (3) well-conditioned, vertebrae and dorsal pelvis not prominent; (4) overconditioned, spine is a continuous column and vertebrae palpable only with firm pressure; and (5) obese, mouse smooth and bulky with bone structure that disappears under flesh and subcutaneous fat (*Ullman-Culleré and Foltz, 1999*).

Clinical score is a sum of seven parameters ranked on the scale of 0 – normal to 4 – severely impaired. The scores were measured by the same observer at all time points in a blinded manner. For the measurement, an animal was transferred to a new empty cage and allowed to acclimate for 30 s. Then, appearance of the body, appearance of the eyes, level of consciousness, and activity were observed. Then, the cage was tapped twice for auditory stimulus, and the animal was touched gently on the back for touch stimulus or gently tipped over if no locomotion was observed. Then, the animal was restrained with a standard hold, allowed to acclimate for 30 s, and its breathing rate and quality were observed and recorded.

## *Listeria* preparation and infection

For all the *Listeria monocytogenes* (Lm) infections, the recombinant Lm strain Lm-OVA, kindly provided by the laboratory of Dr. Hao Shen, was used. Lm were grown to early log phase (OD600 = 0.1) in brain heart infusion media (BHI, BD, cat. 237200) at 37°C, washed in PBS (Corning, cat. 21030CV) and diluted to $10^4$ CFUs per 100 µl of PBS using our observation that 1 OD unit = $3 \times 10^9$ CFUs. The CFU count was confirmed after infection by plating an aliquot of diluted Lm on BHI plate and incubating overnight at 37°C. Experimental animals were temporarily sedated using Isoflurane (Med Vet International, cat. RXISO-250). 100 µl of Lm suspension was injected retro-orbitally per mouse. Mice were observed for full anesthesia recovery. All mice were observed for signs of sickness during infection progression. Any mice that passed before the endpoint were tested for CFUs in their liver and recorded as 'succumbed to infection' if CFUs were detected or 'censored' if CFUs were not detected. For 10 40RD mice and 10 age-matched AL controls, clinical scores were recorded daily at the time of infection and for 14 days following infection.

## Tissue collection and processing

At the end of the experiment, mice were euthanized by carbon dioxide and subsequent cervical dislocation. Either all or a subset of the following tissues were collected, depending on the type of experiment: spleen, bone marrow, thymus, liver, kidney, and inguinal lymph nodes. For steady-state assessment, all the tissues were weighed and photographed except for bone marrow before being processed. Liver and kidney were then disposed of, and only spleen, thymus, and bone marrow were processed. For the infection model, only spleen and bone marrow were processed, and liver was used for CFU counting described in the next section. Bone marrow was harvested and homogenized by flushing bones with a 25-G needle and 10 ml of cold PBS. Thymus and spleen were harvested in full, homogenized in PBS, and strained through a 70-μm cell filter (Celltreat, cat. 229483). All homogenates were resuspended in 1 ml of ACK buffer (Quality Biological, cat. 118-156-101) for 2 min to perform red blood cell lysis. The reaction was quenched with 10 ml of RPMI 1640 supplemented with 10% FBS (Fisher Scientific, cat. BP9703100), 1% Penicillin–streptomycin (Mediatech, cat. MT30-002-CI), 1% L-glutamate (Mediatech, cat. 1249), 10 mM HEPES (Invitrogen, cat. 15630080), 1 mM sodium pyruvate (CCS, cat. 1175), and 0.1% 2-Mercaptoethanol (Life technologies, cat. 21985023). Cells then were counted using a hemocytometer and Trypan Blue (Corning, cat. 25-900-CI).

## Liver processing and CFU count

Livers were harvested in a blinded manner from Lm-infected mice at days 5, 8, and 14 post-infection as described above. Left lateral lobes were collected and weighed. Then, the lobes were homogenized in 1 ml of PBS using the Bead Lysis Kit (Next Advance, cat. PINK5E100) and Next Advance Bullet Blender 5E machine for 10 min at power 8. The homogenates were serially diluted in PBS between 1:5 and 1:200,000 using serial dilution. The dilutions were plated on BHI plates and incubated at 37°C overnight. A plate with a countable number of CFUs (20–100 colonies) was picked per sample. The CFUs were counted by eye, and using the weight of the lobe and dilution factor, CFUs/g of liver were calculated.

## Flow cytometry and cytokine staining

For surface staining, $4 \times 10^6$ cells from spleen, thymus, and bone marrow homogenates were washed with PBS and stained with a LIVE/DEAD Fixable Aqua Dead Cell Stain (Fisher Scientific, cat. L34966) for 15 min at 4°C. Then, cells were washed and stained for 20 min at 4°C in 2% Rat Serum (StemCell, cat. 13551) and PBS with the following antibodies based on the type of the panel (details below).

### Innate immune cells

anti-MHC II (eBioscience M5/114.15.2, cat. 48-5321-80, 1:200), anti-CD11b (eBioscience M1/70, cat. 64-0112-82), anti-CD19 (BioLegend 6D5, cat. 115506), anti-Ly6G (BioLegend 1A8, cat. 127616), anti-Ly6C (eBioscience HK1.4, cat. 17-5932-82), anti-Siglec F (eBioscience 1RNM44N, cat. 127616), anti-TCRβ (eBioscience H57-597, cat. 61-5961-82), anti-CD11c (eBioscience N418, cat. 61-5961-82), anti-F4/80 (eBioscience BM8, cat. 17-4801-82, 1:200), and anti-NK1.1 (eBioscience PK136, cat. 47-5941-82).

### Myelopoiesis

anti-C16/32 (eBioscience 93, cat. 14-0161-81), anti-CD3 (eBioscience 145-2c11, cat. 11-0031-85), anti-CD4 (BioLegend GK1.5, cat. 100406), anti-CD8 (Invitrogen 53-6.7, cat. MA1-10303), anti-CD19 (BioLegend 6D5, cat. 115506), anti-CD11b (BioLegend M1/70, cat. 101206), anti-CD11c (eBioscience N418, cat. 11-0114-85), anti-Ter119 (eBioscience TER-110, cat. 11-5921-85), anti-Ly6G/C (eBioscience RB6-8C5, cat. 11-5931-85), anti-B220 (eBioscience RA3-6B2, cat. 11-0452-85), anti-NK1.1 (eBioscience PK136, cat. 11-5941-85), anti-Sca1 (eBioscience D7, cat. 45-5981-82), CD150 (eBioscience mShed 150, cat. 12-1502-82), CD117 (eBioscience 2B8, cat. 61-1171-82), CD105 (eBioscience MJ7/18, cat. 25-1051-82), CD135 (eBioscience A2F10, cat. 17-1351-82), and CD48 (BD HM48.1, cat. 561242, 1:200).

## Effector T cells

anti-CD4 (eBioscience GK1.5, cat. 64-0041-82), anti-CD8 (Invitrogen 53-6.7, cat. MA1-10303), anti-CD44 (eBioscience IM7, cat. 45-0441-82), anti-CD127 (eBioscience A7R34, cat. 12-1271-82, 1:200), and anti-KLRG1 (eBioscience 2F1, cat. 25-5893-82).

After staining, cells were washed with PBS. If cells were extracted from infected mice, they were also fixed for 30 min at 4°C using eBiosciences Intracellular Fixation Buffer (Termo, cat. 88-8824-00).

For intracellular cytokine staining, $4 \times 10^6$ splenocytes from infected mice were plated in 100 µl of supplemented RPMI 1640 described above in U-bottom tissue culture treated 96-well plates (Celltreat, cat. 229190). Cells were stimulated with 2 µg/ml of SIINFEKL peptide (Anaspec, cat. AS-60193-5) and 1 mg/ml of Brefeldin A (Invitrogen, cat. inh-bfa) added 1 hr after the peptide for a total of 6 hr at 37°C. After incubation, cells were washed with PBS and stained with L/D aqua dye for 15 min at 4°C. Then, cells were washed and stained for 20 min at 4°C in 2% Rat Serum and PBS with the following antibodies: anti-CD4 (eBioscience GK1.5, cat. 64-0041-82), anti-CD8 (Invitrogen 53-6.7, cat. MA1-10303), and anti-CD44 (eBioscience IM7, cat. 45-0441-82). After staining, cells were washed with PBS and fixed for 30 min at 4°C using eBiosciences Intracellular Fixation Buffer. Cells then were washed using eBiosciences Intracellular Permeabilization Buffer (Termo, cat. 88-8824-00) and stained for 30 min at room temperature in 2% Rat Serum and Permeabilization Buffer with the following antibodies: anti-TNFα (eBioscience MP6-XT22, cat. 12-7321-82), anti-IL-2 (eBioscience MQ1-17H12, cat. 25-7029-42, 1:200), and anti-IFNγ (eBioscience XMG1.2, cat. 17-7311-82).

All antibodies used at 1:300 dilution unless otherwise specified. Regardless of staining, all cells were washed with PBS at the end of the protocol and resuspended in PBS to perform flow cytometry. Stained samples were analyzed on a 4-laser CytoFlex by Beckman Coulter using an automated plate reader. Data were analyzed using FlowJo 10.5.3 software.

## Statistics and reproducibility

All experiments were repeated at least three independent times. Data were represented either as a mean ± SD or SEM. Throughout the manuscript, no explicit power analysis was used, but group size was based on previous studies using similar approaches. Blinding was used for CFU measurements, body conditioning score, and clinical score. For measurements over time within the same experimental cohort, statistical analysis was done via simple linear regression comparing slopes or elevation. For survival curves, the log-rank test was performed. For all other pairwise comparisons, the Mann–Whitney test was performed. For all other measurements, statistical analysis was done via non-parametric one-way (Kruskal–Wallis) or two-way (mixed-effect) ANOVA depending on the number of variables. p-values are recorded on the graphs as follows: **** for $p < 0.0001$; *** for $p < 0.001$; ** for $p < 0.01$; * for $p \leq 0.05$; ns for $p > 0.05$. All statistical analysis was performed using GraphPad Prism 9.2.0 software.

## Acknowledgements

We thank the Children's Hospital Flow Cytometry Core for providing support and instrumentation; Drs. Sarah Henrickson, Kelly Jurado, Sunny Shin, and Paula Oliver for feedback and thoughtful discussion; and all members of the Bailis laboratory for providing feedback and support. National Institutes of Health grant R35GM138085 (WB), Paul G Allen Family Foundation Frontiers Group Distinguished Investigator Award (FCB and WB), and Immunobiology of Normal and Neoplastic Lymphocytes Training Grant T32CA009140 (KR).

## Additional information

### Funding

| Funder | Grant reference number | Author |
| --- | --- | --- |
| National Institute of General Medical Sciences | R35GM138085 | Will Bailis |

| Funder | Grant reference number | Author |
| --- | --- | --- |
| Paul G. Allen Frontiers Group | | F Chris Bennett<br>Will Bailis |
| National Cancer Institute | T32CA009140 | Kelly Rome |
| Ludwig Cancer Research | | Will Bailis |

The funders had no role in study design, data collection, and interpretation, or the decision to submit the work for publication.

## Author contributions

Alisa Sukhina, Conceptualization, Data curation, Formal analysis, Validation, Investigation, Visualization, Methodology, Writing – original draft; Clemence Queriault, Saptarshi Roy, Data curation, Formal analysis, Validation, Investigation; Elise Hall, Formal analysis, Validation, Investigation; Kelly Rome, Formal analysis, Investigation; Muskaan Aggarwal, Elizabeth Nunn, Ashley Weiss, Validation, Investigation; Janet Nguyen, Investigation; F Chris Bennett, Resources; Will Bailis, Conceptualization, Resources, Formal analysis, Supervision, Funding acquisition, Visualization, Methodology, Writing – original draft, Project administration, Writing – review and editing

## Author ORCIDs

F Chris Bennett  https://orcid.org/0000-0003-2570-0620
Will Bailis  https://orcid.org/0000-0001-9420-6250

## Ethics

This study was performed in strict accordance with the recommendations in the Guide for the Care and Use of Laboratory Animals of the National Institutes of Health and with PHS-approved animal welfare (assurance number D16-00280). All of the animal were handled according to approved Institutional Animal Care and Use Committee (IACUC) protocols (#001325) of the Children's Hospital of Philadelphia and issues of pain, distress, and injury were addressed throughout this process.

Reviewer #2 (Public review): https://doi.org/10.7554/eLife.101670.3.sa1
Reviewer #3 (Public review): https://doi.org/10.7554/eLife.101670.3.sa2
Author response https://doi.org/10.7554/eLife.101670.3.sa3

# Additional files

## Supplementary files
MDAR checklist

## Data availability

All data generated or analyzed during this study are included in the manuscript and supporting files; source data files have been provided for all figures. No new datasets were generated or previously published datasets analysed as part of this study.

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
