## [Editor Report · eLife Assessment]

This **important** work advances our understanding of the impact of malnutrition on hematopoiesis and subsequently infection susceptibility. Support for the overall claims is **convincing** in some respects and incomplete in terms of identifying mechanism as highlighted by reviewers. This work will be of general interest to those in the fields of hematopoiesis, malnutrition, and dietary influence on immunity.

---

## [Referee Report · Reviewer #2 (Public review)]

Summary:

Sukhina et al. uses a chronic murine dietary restriction model to investigate the cellular mechanisms underlying nutritionally acquired immunodeficiency as well as the consequences of a refeeding intervention. The authors report a substantial impact of undernutrition to the myeloid compartment, which is not rescued by refeeding despite rescue of other phenotypes including lymphocyte levels, and which is associated with maintained partial susceptibility to bacterial infection.

Strengths:

Overall, this is a nicely executed study with an appropriate number of mice, robust phenotypes, and interesting conclusions, and the text is very well written. The authors' conclusions are generally well-supported by their data.

Weaknesses:

There is little evaluation of known critical drivers of myelopoiesis (e.g. PMID 20535209, 26072330, 29218601) over the course of the 40% diet, which would be of interest with regard to comparing this chronic model to other more short-term models of undernutrition.

Further, the microbiota, well-established to be regulated by undernutrition (e.g. PMID 22674549, 27339978, etc.), and also well-established to be a critical regulator of hematopoiesis/myelopoiesis (e.g. PMID 27879260, 27799160, etc.), should be studied in any future explorations using this model.

The authors have recognized these limitations to the study in their discussion.

---

## [Referee Report · Reviewer #3 (Public review)]

This communication from Sukhina et al argues that a period of malnutrition (modeled by caloric restriction) causes lasting immune deficiencies (myelopoesis) not rescued by re-feeding. This is a potentially important paper exploring the effects of malnutrition on immunity, which is a clinically important topic. The revised study adds some details with respect to kinetics of immune compartment and body weight changes, but most aspects raised by the referees were deferred experimentally. Several textual changes have been made to avoid over-interpreting their data. My overall assessment of this revised study is similar to my impression before, which is that while the observations are interesting, there is both a lack of mechanistic understanding of the phenomena and a lack of resolution/detail about the phenomena itself.

---

## [Author Response]

The following is the authors’ response to the original reviews

**eLife Assessment**
This important work advances our understanding of the impact of malnutrition on hematopoiesis and subsequently infection susceptibility. Support for the overall claims is convincing in some respects and incomplete in others as highlighted by reviewers. This work will be of general interest to those in the fields of hematopoiesis, malnutrition, and dietary influence on immunity.

We would like to thank the editors for agreeing to review our work at eLife. We greatly appreciate them assessing this study as important and of general interest to multiple fields, as well as the opportunity to respond to reviewer comments. Please find our responses to each reviewer below.

**Public Reviews:**

**Reviewer #1 (Public review):**
Summary:In this study, the authors used a chronic murine dietary restriction model to study the effects of chronic malnutrition on controls of bacterial infection and overall immunity, including cellularity and functions of different immune cell types. They further attempted to determine whether refeeding can revert the infection susceptibility and immunodeficiency. Although refeeding here improves anthropometric deficits, the authors of this study show that this is insufficient to recover the impairments across the immune cell compartments.Strengths:The manuscript is well-written and conceived around a valid scientific question. The data supports the idea that malnutrition contributes to infection susceptibility and causes some immunological changes. The malnourished mouse model also displayed growth and development delays. The work's significance is well justified. Immunological studies in the malnourished cohort (human and mice) are scarce, so this could add valuable information.Weaknesses:The assays on myeloid cells are limited, and the study is descriptive and overstated. The authors claim that "this work identifies a novel cellular link between prior nutritional state and immunocompetency, highlighting dysregulated myelopoiesis as a major." However, after reviewing the entire manuscript, I found no cellular mechanism defining the link between nutritional state and immunocompetency.

We thank the reviewer for deeming our work significant and noting the importance of the study. We appreciate the referee’s point regarding the lack of specific cellular functional data for innate immune cells and have modified the conclusions stated in text to more accurately reflect the results presented.

**Reviewer #2 (Public review):**
Summary:Sukhina et al. use a chronic murine dietary restriction model to investigate the cellular mechanisms underlying nutritionally acquired immunodeficiency as well as the consequences of a refeeding intervention. The authors report a substantial impact of undernutrition on the myeloid compartment, which is not rescued by refeeding despite rescue of other phenotypes including lymphocyte levels, and which is associated with maintained partial susceptibility to bacterial infection.Strengths:Overall, this is a nicely executed study with appropriate numbers of mice, robust phenotypes, and interesting conclusions, and the text is very well-written. The authors' conclusions are generally well-supported by their data.Weaknesses:There is little evaluation of known critical drivers of myelopoiesis (e.g. PMID 20535209, 26072330, 29218601) over the course of the 40% diet, which would be of interest with regard to comparing this chronic model to other more short-term models of undernutrition.Further, the microbiota, which is well-established to be regulated by undernutrition (e.g. PMID 22674549, 27339978, etc.), and also well-established to be a critical regulator of hematopoiesis/myelopoiesis (e.g. PMID 27879260, 27799160, etc.), is completely ignored here.

We thank the reviewer for agreeing that the data presented support the stated conclusions and noting the experimental rigor. The referee highlights two important areas for future mechanistic investigation that we agree are of great importance and relevant to the submitted study. We have included further discussion of the potential role cytokines and the microbiota might play in our model.

**Reviewer #3 (Public review):**
Summary:Sukhina et al are trying to understand the impacts of malnutrition on immunity. They model malnutrition with a diet switch from ad libitum to 40% caloric restriction (CR) in post-weaned mice. They test impacts on immune function with listeriosis. They then test whether re-feeding corrects these defects and find aspects of emergency myelopoiesis that remain defective after a precedent period of 40% CR. Overall, this is a very interesting observational study on the impacts of sudden prolonged exposure to less caloric intake.Strengths:The study is rigorously done. The observation of lasting defects after a bout of 40% CR is quite interesting. Overall, I think the topic and findings are of interest.Weaknesses:While the observations are interesting, in this reviewer's opinion, there is both a lack of mechanistic understanding of the phenomena and also some lack of resolution/detail about the phenomena itself. Addressing the following major issues would be helpful towards aspects of both:(1) Is it calories, per se, or macro/micronutrients that drive these phenotypes observed with 40% CR. At the least, I would want to see isocaloric diets (primarily protein, fat, or carbs) and then some of the same readouts after 40% CR. Ie does low energy with relatively more eg protein prevent immunosuppression (as is commonly suggested)? Micronutrients would be harder to test experimentally and may be out of the scope of this study. However, it is worth noting that many of the malnutrition-associated diseases are micronutrient deficiencies.(2) Is immunosuppression a function of a certain weight loss threshold? Or something else? Some idea of either the tempo of immunosuppression (happens at 1, in which weight loss is detected; vs 2-3, when body length and condition appear to diverge; or 5 weeks), or grade of CR (40% vs 60% vs 80%) would be helpful since the mechanism of immunosuppression overall is unclear (but nailing it may be beyond the scope of this communication).(3) Does an obese mouse that gets 40% CR also become immunodeficient? As it stands, this ad libitum  40% CR model perhaps best models problems in the industrial world (as opposed to always being 40% CR from weaning, as might be more common in the developing world), and so modeling an obese person losing a lot of weight from CR (like would be achieved with GLP-1 drugs now) would be valuable to understanding generalizability.(4) Generalizing this phenomenon as "bacterial" with listeriosis, which is more like a virus in many ways (intracellular phase, requires type I IFN, etc.) and cannot be given by the natural route of infection in mice, may not be most accurate. I would want to see an experiment with *E. coli*, or some other bacteria, to test the statement of generalizability (ie is it bacteria, or type I IFN-pathway dominant infections, like viruses). If this is unique listeriosis, it doesn't undermine the story as it is at all, but it would just require some word-smithing.(5) Previous reports (which the authors cite) implicate Leptin, the levels of which scale with fat mass, as "permissive" of a larger immune compartment (immune compartment as "luxury function" idea). Is their phenotype also leptin-mediated (ie leptin AAV)?(6) The inability of re-feeding to "rescue" the myeloid compartment is really interesting. Can the authors do a bone marrow transplantation (CRad libitum) to test if this effect is intrinsic to the CR-experienced bone marrow?(7) Is the defect in emergency myelopoiesis a defect in G-CSF? Ie if the authors injected G-CSF in CR animals, do they equivalently mobilize neutrophils? Does G-CSF supplementation (as one does in humans) rescue host defense against Listeria in the CR or re-feeding paradigms?

We thank the reviewer for considering our work of interest and noting the rigor with which it was conducted. The referee raises several excellent mechanistic hypotheses and follow-up studies to perform. We agree that defining the specific dietary deficiency driving the phenotypes is of great interest. The relative contribution of calories versus macro- and micronutrients is an area we are interested in exploring in future studies, especially given the literature on the role of micronutrients in malnutrition driven wasting as the referee notes. We also agree that it will be key to determine whether non-hematopoietic cells contribute as well as the role of soluble factors such G-CSF and Leptin in mediating the immunodeficiency all warrant further study. Likewise, it will be important to evaluate how malnutrition impacts other models of infection to determine how generalizable these phenomena are. We have added these points to the discussion section as limitations of this study.

Regarding how the phenotypes correspond to the timing of the immunosuppression relative to weight loss, we have performed new kinetics studies to provide some insight into this area. We now find that neutropenia in peripheral blood can be detected after as little as one week of dietary restriction, with neutropenia continuing to decline after prolonged restriction. These findings indicate that the impact on myeloid cell production are indeed rapid and proceed maximum weight loss, though the severity of these phenotypes does increase as malnutrition persists. We wholeheartedly agree with the reviewer that it will be interesting to explore whether starting weight impacts these phenotypes and whether similar findings can be made in obese animals as they are treated for weight loss.

**Recommendations for the authors:**

**Reviewer #1 (Recommendations for the authors):**
In this study, the authors used a chronic murine dietary restriction model to study the effects of chronic malnutrition on controls of bacterial infection and overall immunity, including cellularity and functions of different immune cell types. They further attempted to determine whether refeeding can revert the infection susceptibility and immunodeficiency. Although refeeding here improves anthropometric deficits, the authors of this study show that this is insufficient to recover the impairments across the immune cell compartments. The authors claim that "this work identifies a novel cellular link between prior nutritional state and immunocompetency, highlighting dysregulated myelopoiesis as a major." However, after reviewing the entire manuscript, I could not find any cellular mechanism defining the link between nutritional state and immunocompetency. The assays on myeloid cells are limited, and the study is descriptive and overstated.Major concerns:(1) Malnutrition has entirely different effects on adults and children. In this study, 6-8 weeks old C57/Bl6 mice were used that mimic adult malnutrition. I do not understand then why the refeeding strategy for inpatient treatment of severely malnourished children was utilized here.(2) Figure 1g shows BM cellularity is reduced, but the authors claim otherwise in the text.(3) What is the basis of the body condition score in Figure 1d? It will be good to have it in the supplement.(4) Listeria monocytogenes cause systemic infection, so bioload was not determined in tissues beyond the liver.(5) Figure 3; T cell functional assays were limited to CD8 T cells and lymphocytes isolated from the spleen.(6) Why was peripheral cell count not considered? Discrepancies exist with the absolute cell number and relative abundance data, except for the neutrophil and monocyte data, which makes the data difficult to interpret. For example, for B cells, CD4 and CD8 cells.(7) Also, if mice exhibit thymic atrophy, why does % abundance data show otherwise? Overall, the data is confusing to interpret.(8) No functional tests for neutrophil or monocyte function exist to explain the higher bacterial burden in the liver or to connect the numbers with the overall pathogen loadThe rationale for examining both innate and adaptive immunity is not clear-it is even more unclear since the exact timelines for examining both innate and adaptive immunity (D0 and D5) were used.(9) Figure 2e doesn't make sense - why is spleen cellularity measured when bacterial load is measured in the liver?(10) Although it is claimed that emergency myelopoiesis is affected, no specific marker for emergency myelopoiesis other than cell numbers was studied.(11) I suggest including neutrophil effector functions and looking for real markers of granulopoiesis, such as Cebp-b. Since the authors attempted to examine the entirety of immune responses, it is better to measure cell abundance, types, and functions beyond the spleen. Consider the systemic spread of m while measuring bioload.(12) Minor grammatical errors - please re-read the entire text and correct grammatical errors to improve the flow of the text.(13) Sample size details missing(14) Be clear on which marks were used to identify monocytes. Using just CD11b and Ly6G is insufficient for neutrophil quantification.(15) Also, instead of saying "undernourished patients," say "patients with undernutrition" - change throughout the text. I would recommend numbering citations (as is done for Nature citations) to ease in following the text, as there are areas when there are more than ten citations with author names.(16) No line numbers are provided(17) Abstract- What does accelerated contraction mean?- "In" is repeated in a sentence- Be clear that the study is done in a mouse model - saying just "animals" is not sufficient- Indicate how malnutrition is induced in these mice(18) Introduction- "restriction," "immune organs," - what is this referring to?- You mention lymphoid tissue and innate and adaptive immunity, which doesn't make sense.Please correct this.- You mention a lot of lymphoid tissues, i.e. lymphoid mass gain, but how about the bone marrow and spleen, which are responsible for most innate immune compartments?(19) Resultsa) Figure 1- Why 40% reduced diet?- It would be interesting to report if the organs are smaller relative to body weight. It makes sense that the organ weight is lower in the 40RD mice, especially since they are smaller, so the novelty of this data is not apparent (Figure 1f).- You say, "We observed a corresponding reduction in the cellularity of the spleen and thymus, while the cellularity of the bone marrow was unaffected (Fig. 1g)." however, your BM data is significant, so this statement doesn't reflect the data you present, please correct.b) Figure 2- Figure 2d - what tissue is this from, mentioned in the figure? And measure cellularity there. The rationale for why you look only at the spleen here is weak. Also, we would benefit from including the groups without infection here for comparison purposes.c) Figure 3- The rationale for why you further looked at T cells is weak, mainly because of the following sentence. "Despite this overall loss in lymphocyte number, the relative frequency of each population was either unchanged or elevated, indicating that while malnutrition leads to a global reduction in immune cell numbers, lymphocytes are less impacted than other immune cell populations (Supplemental 1)." Please explain in the main text.d) Figure 4- You say the peak of the adaptive immune response, but you never looked at the peak of adaptive immune - when is this? If you have the data, please show it. You also only show d0 and d5 post-infection data for adaptive immunity, so I am unsure where this statement comes from.- How did you identify neutrophils and monocytes through flow cytometry? Indicate the markers used. Also, your text does not match your data; please correct it. i.e. monocyte numbers reduced, and relative abundance increased, but your text doesn't say this.- Show the flow graph first then, followed by the quantification.- The study would benefit from examining markers of emergency myelopoiesis such as Cebpb through qPCR.- Although the number of neutrophils is lower in the BM and spleen, how does this relate to increased bacterial load in the liver? This is especially true since you did not quantify neutrophil numbers in the liver.e) Figure 6- Some figures are incorrectly labelled.- For the refeeding data, also include the data from the 40RD group to compare the level of recovery in the outcome measures.(20) Discussion- You claim that monocytes are reduced to the same extent as neutrophils, but this is not true.Please correct.- Indicate some limitations of your work.

We thank the reviewer for offering these recommendations and the constructive comments.

Several comments raised concerns over the rationale or reasoning behind aspects of the experimental design or the data presented, which we would like to clarify:

• Regarding the refeeding protocol, we apologize for the confusion for the rationale. We based our methodology on the *general* guidelines for refeeding protocols for malnourished people. We elected to increase food intake 10% daily to avoid risk of refeeding syndrome or other complications. Our method is by no means replicates the administration of specific vitamins, minerals, electrolytes, nor precise caloric content as would be given to a human patient. The citation provided offers information from the WHO regarding the complications that can arise during refeeding syndrome, which while it is from a document on pediatric care, we did not mean to imply that our method modeled refeeding intervention for children. We have modified the text to avoid this confusion.

• The reviewer requested more clarity on why we studied both the innate and adaptive immune system as well as why we chose the time points studied. As referenced in the manuscript, prior work has observed that caloric restriction, fasting, and malnutrition all can impact the adaptive immune system. Given these previous findings, we felt it important to evaluate how malnutrition affected adaptive immune cell populations in our model. To this end, we provide data tracking the course of T-cell responses from the start of infection through day 14 at the time that the response undergoes contraction. However, since we find that bacterial burden is not properly controlled at earlier time points (day 5), when it is understood the innate immune system is more critical for mediating pathogen clearance, we elected to better characterize the effect malnutrition had on innate immune populations, something less well described in the literature. As phenotypes both in bacterial burden and within innate immune populations were observable as early as day 5, we chose to focus on that time point rather than later time points when readouts could be further confounded by secondary or compounding effects by the lack of early control of infection. We have tried to make this rationale clear in the text and have made changes to further emphasize this reasoning.

• The reviewer also requested an explaination over why bacterial burden was measured in the liver and the immune response was measured in the spleen. While the reviewer is correct that our model is a systemic infection, it is well appreciated that bacteria rapidly disseminate to the liver and spleen and these organs serve as major sites of infection. Given the central role the spleen plays in organizing both the innate and adaptive immune response in this model, it is common practice in the field to phenotype immune cell populations in the spleen, while using the liver to quantify bacterial burden (see PMID: 37773751 as one example of many). We acknowledge this does not provide the full scope of bacterial infection or the immune response in every potentially affected tissue, but nonetheless believe the interpretation that malnourished and previously malnourished animals do not properly control infection and their immune responses are blunted compared to controls still stands.

The reviewer raised several points about di3erences in the results for cell frequency and absolute number and why these may deviate in some circumstances. For example, the reviewer notes that we observe thymic atrophy yet the frequency of peripheral T-cells does not decline. It should be noted that absolute number can change when frequency does not and vice versa, due to changes in other cell types within the studied population of cells. As in the case of peripheral lymphocytes in our study, the frequency can stay the same or even increase when the absolute number declines (Supplemental 1). This can occur if other populations of cells decrease further, which is indeed the case as the loss of myeloid cells is greater than that of lymphocytes. Hence, we find that the frequency of T and B cells is unchanged or elevated, despite the loss in absolute number of peripheral cell, which is our stated interpretation. We believe this is consistent with our overall observations and is why it is important to report both frequency and absolute number, as we have done.

We have made the requested changes to the text to address the reviewers concerns as noted to improve clarity and accuracy for the description of experiments, results, and overall conclusions drawn in the manuscript. We have also included a discussion of the limitations of our work as well as additional areas for future investigation that remain open.

**Reviewer #2 (Recommendations for the authors):**
Regarding the known drivers of myelopoiesis, can the authors quantify circulating levels of relevant immune cytokines (e.g. type I and II IFNs, GM-CSF, etc.)?Regarding the microbiota (point #2), how dramatically does this undernutrition modulate the microbiota both in terms of absolute load and community composition, and how effectively/quickly is this rescued by refeeding?

We thank the reviewer for raising these recommendations. We agree that the role of circulating factors like cytokines and growth factors in contributing to the defects in myelopoiesis is of interest and is the focus of future work. Similarly, the impact of malnutrition on the microbiota is of great interest and has been evaluated by other groups in separate studies. How the known impact of malnutrition on the microbiota affects the phenotypes we observe in myelopoiesis is unclear and warrants future investigation. We have added these points to the discussion section as limitations of this study.